# Key Considerations in Assessing the Safety and Performance of Camera-Based Mirror Systems

**Amy Moore ***, **Jinghui Yuan**, **Shiqi (Shawn) Ou**, **Jackeline Rios Torres**, **Vivek Sujan** and **Adam Siekmann**

Oak Ridge National Laboratory, Oak Ridge, TN 37830, USA; yuanj@ornl.gov (J.Y.); ous1@ornl.gov (S.O.); riostorresj@ornl.gov (J.R.T.); sujanva@ornl.gov (V.S.); siekmanna@ornl.gov (A.S.)
* Correspondence: mooream@ornl.gov

**Abstract:** Camera-based mirror systems (CBMSs) are a relatively new technology in the automotive industry, and much of the United States' medium- and heavy-duty commercial fleet has been reluctant to convert from standard glass, or "west coast", mirrors to CBMSs. CBMSs have the potential to reduce the number of truck and passenger vehicle incidents, improving overall fleet safety. CBMSs also have the potential to improve operational efficiency by improving aerodynamics and reducing drag, resulting in better fuel economy, and improving maneuverability. Improvements in overall safety are also possible; the field of view for the driver is potentially 360° with the addition of trailer cameras, allowing for visibility of the rear of the trailer and the front of the truck. These potential improvements seem promising, but the literature on driver surveys clearly shows that there is reluctance to adopt this technology for many reasons. Additionally, more robust testing in the laboratory and in the field is necessary to determine whether CBMSs are adequate to replace standard mirrors on trucks. This analysis provides an overview of key research questions for CBMS testing based on the current literature on the topic (surveys, standards, and previous testing). The purpose of this analysis is to serve as guidance in developing further testing of CBMSs, especially testing involving human subjects.

**Keywords:** camera-based mirror systems; mirrors; trucks; fleet safety; operational efficiency





## 1. Introduction to Camera-Based Mirror Systems

The use of camera-based mirror systems (CBMSs) in medium- and heavy-duty trucks is a relatively new development. These systems use cameras mounted to the exterior of the truck (Figure 1) and sometimes the trailer, as well as interior monitors mounted inside the truck cabin to project real-time images of the truck's and trailer's surroundings to the driver. CBMSs are becoming increasingly popular in the vehicle industry as a replacement for traditional mirrors. Currently, the typical truck uses standard, or "west coast", glass mirrors on both sides of the vehicle to obtain a field of view that encompasses the side and a portion of the rear of the trailer (Figure 2). However, these glass mirrors leave a large amount of space surrounding the vehicle out of view for the driver, creating blind spots, which unfortunately cause many incidents on roadways [1]. Although many truck drivers and other transportation professionals approve of the use of CBMS technology to avoid a large majority of incidents and improve overall efficiency, the technology is still relatively new, and many drivers are apprehensive about its adoption.

In terms of operational efficiency, CBMSs have the potential to provide an increase in fuel efficiency estimated at up to 2–3% [2]. This increase is largely attributed to a reduction in aerodynamic drag, which occurs because of the size and placement of standard truck mirrors [3]. The additional savings in fuel economy can be attributed to improvements in maneuverability and ease of parking achieved by reducing the occurrence of repeated movements via an aerial view [4]. For a large fleet, these improvements have the potential to lead to significant savings in fuel usage and ultimately overall costs.

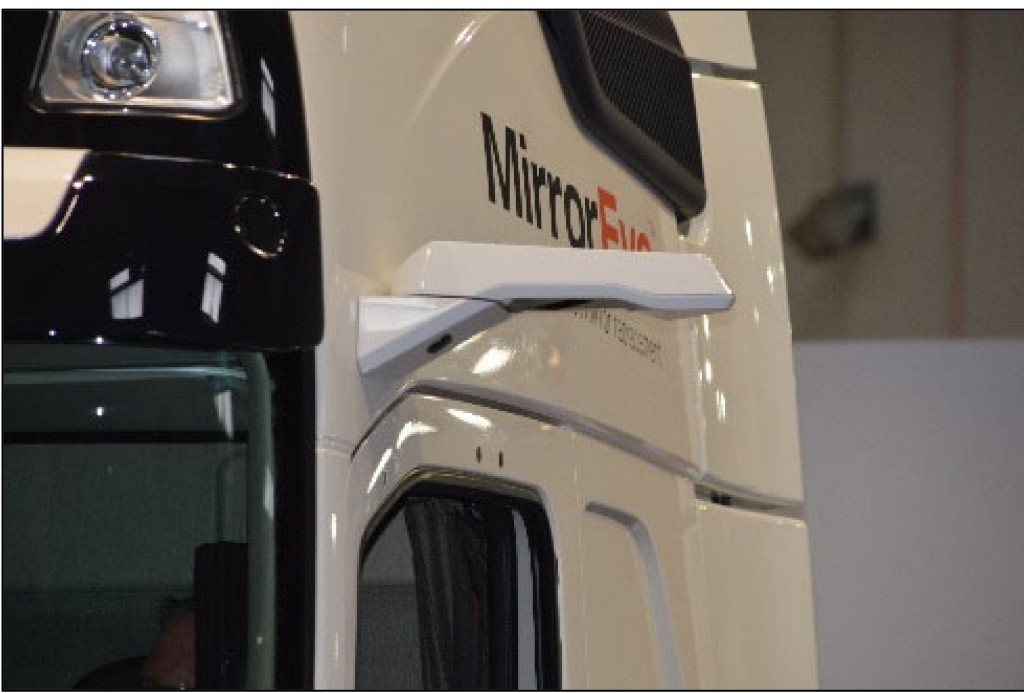

**Figure 1.** MirrorEye camera.

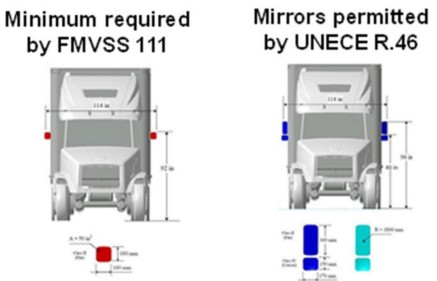

**Figure 2.** Standard and West Coast mirrors.

Potential drawbacks to incorporating CBMSs include cost, maintenance, reliability, and potential system failures. Camera systems on average cost approximately USD 5K per truck installation. For a sizeable fleet, this can be a major expense. Maintenance costs are certainly higher for CBMSs than for mirrors, as they require specialized technicians to make repairs, and downtime for drivers can occur if the needed repairs render the vehicle inoperable or out of service. Without system redundancy and reliability, the potential exists for the system to fail. These are often the major concerns voiced by drivers, as is evident in the following survey evaluations.

The evaluation of the effectiveness of CBMSs in improving safety and efficiency has been a subject of interest for technology manufacturers and especially for policymakers. Several studies [5–7] have elicited the perspectives of commercial truck drivers regarding the use of CBMSs and related technologies. Throughout the literature related to the reluctance to accept this technology, some of the major overarching concerns are redundancy of the system and maintenance of power during electrical issues, relative feelings of safety for both the driver and for other drivers on the road, and driver preparedness and driver qualifications and training [3]. These issues, among many others, warrant both laboratory and field testing to ensure safety and efficiency, and although ongoing surveys and testing are being conducted by both industry and government entities, substantially more research is needed to determine whether CBMSs can adequately replace standard glass mirrors

completely or whether they should be used in conjunction with mirrors. Identifying gaps within the current body of research is crucial to determine the appropriate focus for future research and to help in the formulation of appropriate testing scenarios.

This paper presents an in-depth analysis of the literature on CBMSs as an alternative to standard glass mirrors on medium- and heavy-duty trucks and identifies key research questions that are lacking in the current research on the topic. The analysis performed was performed through an effort by the Federal Motor Carrier Safety Administration with a focus primarily on safety but is applicable to efforts to address the efficient use of resources and to improve fleet operations. This paper is structured as follows: Section 2 contains an overview of overarching driver concerns found throughout previous surveys performed by both industry and government entities. This section also contains the background and assumptions used in a failure mode and effects analysis (FMEA) and the justification for a comprehensive list of key research questions to be examined throughout related studies on the topic. Section 3 reviews existing governmental standards that warrant the use of technologies such as CBMSs. Section 4 provides a thorough literature review detailing methodologies used and outlines potential gaps in the literature. Lastly, Section 5 provides concluding remarks and potential expansions of this work. This work assumes that, although further testing and research are needed, CBMSs have the potential to improve safety for truck drivers. This work is limited because further testing and research are needed, and many of the references were based on driver feedback. It is hoped that this work provides guidance for future studies and justifications for specific testing scenarios.

## 2. Development of Research Questions from Surveys and FMEA

### 2.1. Overview of Initial Driver Responses from Surveys

The National Highway Traffic Safety Administration (NHTSA) has performed and is involved in several ongoing studies examining driver behavior using CBMSs and has attempted to address driver concern with the initial use of these systems. In a 2019 study of driver perception upon the initial use of a CBMS [4], the following responses were noted:

- Camera-based rear visibility system displays will make driving unsafe compared with traditional mirrors.
- Drivers will not be able to easily acclimate to using the visual displays of camera-based rear visibility systems and different display locations (if applicable).
- Camera-based rear visibility systems and related new technology will further remove the human from the driving task.
- Drivers doubt the camera-based rear visibility systems' ability to function reliably and that cameras requiring power can fail unexpectedly and cause a lack of awareness of the drivers' surroundings.
- Drivers suspect that law enforcement would have greater difficulty determining whether camera-based rear visibility systems are working correctly compared with traditional mirrors, for which damage can be easily determined.

In one survey conducted in 2016 [8], drivers reported mixed feelings regarding the use of CBMSs. A summary of these responses is in Table 1.

The responses from these driver surveys reflect common themes throughout the literature and are not atypical for first-time users of this technology. A review of driver responses helped in formulating the list of functions, potential failures, and causes of failures used in the FMEA.

### 2.2. Failure Mode and Effects Analysis

An FMEA is an efficient and thorough way to evaluate functions and many of the potential failures and causes of failures within a system. This process enables the quantification of failures by assigning categorical values to the likelihood and severity of failures incurred. This methodology is commonly used in manufacturing but is also applicable in other areas of research and development. As part of this analysis, an FMEA was performed to consider the primary and secondary functions and the numerous sources of failure in

both glass mirrors and CBMSs and to assign weights to these failures based on potential likelihood, criticality, and severity. The assigned score is based on the product of the assumed likelihood, criticality, and severity scores, with each score given a value between 0 and 10.

**Table 1.** Survey responses from the 2016 study.

| Positive | |
|---|---|
| More information provided about the rear and front of the vehicle (reduces occurrences of blind spots) | Disadvantages of the wide-angle mirror (distorted image, strong curvature) are compensated by the CBMS |
| Shallow learning curve for drivers | Improved aerodynamics |
| Better direct view out of windows | Fuel savings |
| More accurate presentation of surroundings than mirrors (mirrors have strong curvature and distorted view) | Front of the trailer clearly visible on the monitor |
| Camera lens stays clear (less condensate during precipitation events compared with standard mirrors) | No head movement required |
| The system is unfamiliar, but one can get used to it. | |
| **Negative** | |
| Unrealistic quality/distorted images misinform driver of depth/distance and potentially speed | Shadow formation too strong |
| Position of monitor needs to be adjusted for each driver | Objects are displayed smaller on the screen |
| Difficulty maneuvering roundabouts | Display could be larger, especially in the right monitor |
| Difficulty adjusting to sudden, dramatic changes in lighting | Flickering/jittering of the image, especially during engine start and turns |
| Angle of lens cover should be adjusted to prevent glare/reflection | Issues for farsighted drivers |
| Redundancy; electromagnetic radiation compatibility | Position of the right monitor is too far; objects are even more difficult to recognize. |
| Heating of the unit should be available to prevent icing in low temperatures and snow | Reduced feeling of safety in comparison to mirrors |
| Prevention of time delay and image loss is required | Dust and fingerprints visible and distracting |
| Poor contrast/color representation | |

Figure 3 clearly shows that glass mirrors, being analog and relatively low-tech compared to CBMSs, have fewer primary and secondary functions than CBMSs. Notably, there are numerous instances of nonapplicable scenarios. It is also noteworthy that the only function that has the potential for a variety of occurrences related to the mirror failing to provide an adequate view of the surroundings for the driver was related to the mirror becoming dirty, fogged over, or covered with ice. Lastly, all boxes in Figure 3 with scores are shades of green, which were used to represent lower-level (less than a score of 100) risks in this FMEA.

| Glass mirror | -No Function | -Stops Functioning | -Unrequested Function | -Function Stuck | -Excessive Function | -Partial Function | -Functions Early | -Functions Late | -Function Applies Too Long | -Function Applies Too Short | -Function is Delayed | -Inverse Function | -Erratic or Intermittent Function | -Function is Uneven |
|---|---|---|---|---|---|---|---|---|---|---|---|---|---|---|
| mirror/reflect and magnify rear image during forward driving (primary) | 51 | 73.7 | N/A | N/A | N/A | 92.1 | N/A | N/A | N/A | N/A | N/A | N/A | N/A | N/A |
| mirror/reflect and magnify rear image during stationary power off or backing up driving (primary) | 42 | 60.7 | N/A | N/A | N/A | 78.2 | N/A | N/A | N/A | N/A | N/A | N/A | N/A | N/A |
| heating/defrosting function (secondary) | 68 | 90.7 | 51 | 51 | 51 | 74.7 | 51 | 74.7 | 51 | 74.7 | 74.7 | 51 | 74.7 | 74.7 |
| Anti-glare feature using film coating (secondary) | 38.5 | 38.5 | N/A | N/A | N/A | 60.8 | N/A | N/A | N/A | N/A | N/A | N/A | N/A | N/A |
| mirror angle set and maintained appropriately for clear rear view visibility (secondary) | 50 | 64 | N/A | 64 | N/A | N/A | N/A | N/A | N/A | N/A | N/A | N/A | N/A | N/A |
| arm/restraint to hold mirror in place (secondary) | 50 | 64 | N/A | N/A | N/A | 82 | N/A | N/A | N/A | N/A | N/A | N/A | N/A | N/A |

**Figure 3.** Functions and failure modes of glass mirrors.

As shown in Figure 4, CBMSs have many more opportunities for failure than glass mirrors. This is indicated with the chosen color symbology: green, yellow, orange, and red, in the order of increasing score. More functions are inherent to CBMSs, and there are also many more scenarios where failure of these functions can occur because of a variety of causal variables. Many of these failures could be catastrophic because of the severity inherent to these events (e.g., the driver completely losing their ability to see their surroundings due to a failure of the system, resulting in a designated dark orange or red symbology, based on severity).

| FUNCTION \ CBMS FAILURE MODE | -No Function | -Stops Functioning | -Unrequested Function | -Function Stuck | -Excessive Function | -Partial Function | -Functions Early | -Functions Late | -Function Applies Too Long | -Function Applies Too Short | -Function is Delayed | -Inverse Function | -Erratic or Intermittent Function | -Function is Uneven |
|---|---|---|---|---|---|---|---|---|---|---|---|---|---|---|
| camera accepts light rays and presents rear image to driver, with appropriate magnification during forward driving (primary) | 144.6 | 177 | 227.3 | 227.3 | 222.6 | 218.6 | 222.6 | 221.4 | 222.6 | 221.4 | 221.4 | 221.4 | 221.4 | 221.4 |
| camera accepts light rays and presents rear image to driver, with appropriate magnification during stationary power off or backing up driving (primary) | 121.7 | 143.3 | 191.5 | 191.5 | 187.5 | 183.8 | 187.5 | 186.5 | 187.5 | 186.5 | 186.5 | 186.5 | 186.5 | 186.5 |
| heating/defrosting function (secondary) | 74.4 | 74.4 | 74.4 | 74.4 | 74.4 | 74.4 | 74.4 | 74.4 | 74.4 | 74.4 | 74.4 | 74.4 | 74.4 | 74.4 |
| Anti-glare feature using electronic controls (secondary) | 141.4 | 153.8 | 153.8 | 153.8 | 153.8 | 153.8 | 153.8 | 153.8 | 153.8 | 153.8 | 153.8 | 153.8 | 153.8 | 153.8 |
| camera angle set and maintained appropriately for clear rear view visibility (secondary) | 142.6 | 181.9 | 181.9 | 181.9 | N/A | 181.9 | 135.2 | 135.2 | 137.4 | 137.4 | 135.2 | 135.2 | 137.4 | N/A |
| arm/restraint to hold camera in place (secondary) | 115.7 | 115.7 | 115.7 | 115.7 | N/A | 115.7 | N/A | N/A | N/A | N/A | N/A | N/A | 115.7 | 115.7 |
| image zoom (FOV)/pan/tilt/resolution function (secondary) | 141.4 | 153.8 | 153.8 | 153.8 | 153.8 | 153.8 | 153.8 | 153.8 | 153.8 | 153.8 | 153.8 | 153.8 | 153.8 | 153.8 |
| power to system function (primary) | 55.8 | 74.4 | N/A | 45 | 54 | 55.8 | 45 | 37.2 | 45 | 87 | 37.2 | 87 | 55.8 | N/A |
| screen luminance control function for driver comfort (secondary) | 55.8 | 55.8 | 55.8 | 46.8 | 46.8 | 46.8 | N/A | N/A | N/A | N/A | N/A | 55.8 | 55.8 | N/A |
| Maintenance of uniform image/contrast/rendering/grey scale/color to minimize depth/image distortion/frame rate (primary) | 117.8 | 117.8 | 117.8 | 117.8 | N/A | 117.8 | N/A | N/A | N/A | N/A | N/A | N/A | 117.8 | N/A |
| camera accepts light rays and presents 360 deg view image to driver, with appropriate magnification during forward/reverse driving (secondary) | 135 | 135 | 124 | 124 | 124 | 124 | 124 | 124 | 124 | 124 | 124 | 128.9 | 124 | N/A |

**Figure 4.** Functions and failure modes of CBMSs.

In Figure 5, a nine-box plot shows varying levels of criticality and difficulty in resolving failures of functions in glass mirrors and CBMSs. For glass mirrors, most failures are found in Box 1, which represents failures that are low in criticality and low in difficulty to resolve. Although many of the failures of CBMSs are also found in Box 1, many more are found in Boxes 4, 5, and 6, which represent higher criticality and greater difficulty to resolve. This observation raises concerns that must be addressed through further research and testing.

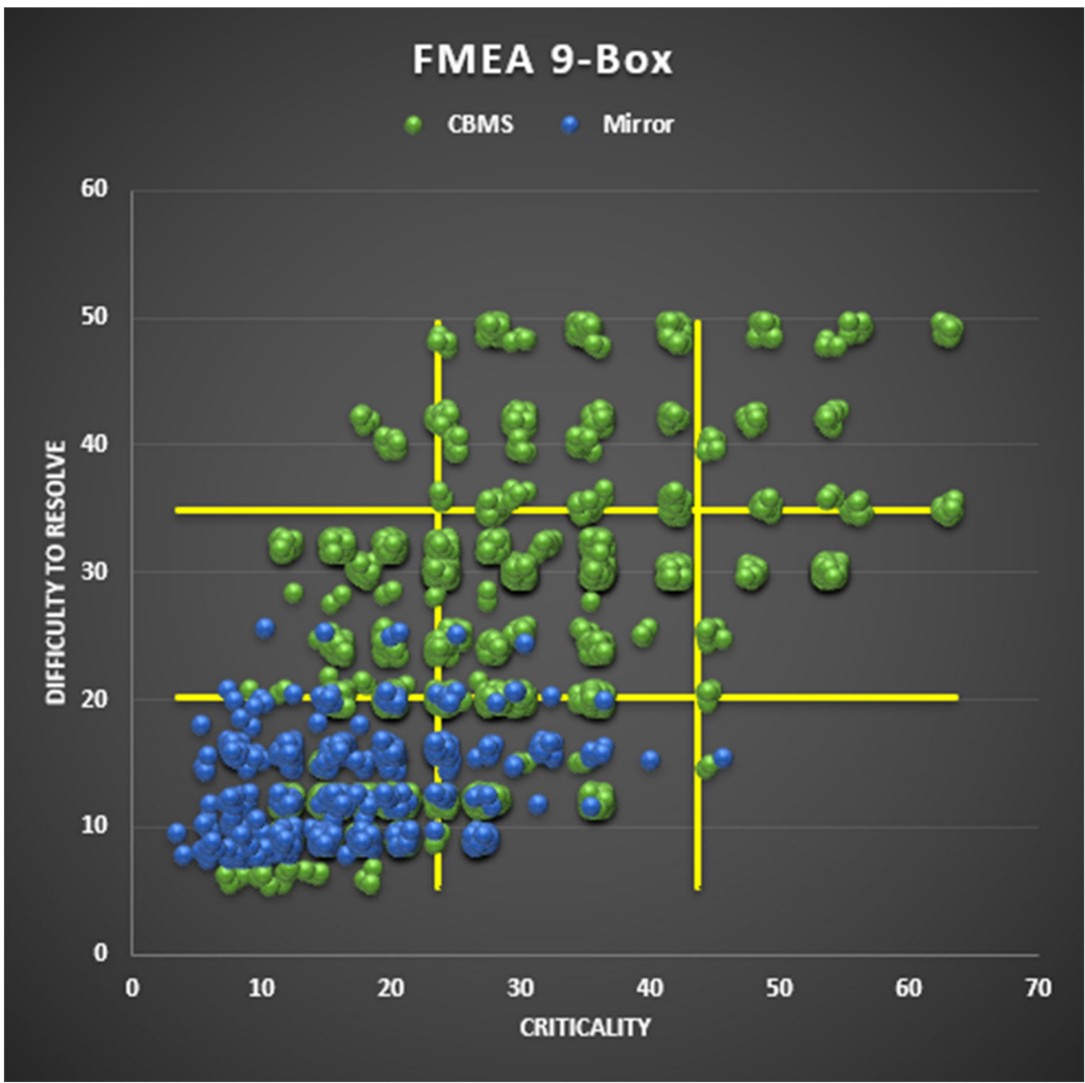

**Figure 5.** Nine-box plot showing levels of criticality of and difficulty in resolving failures.

### 2.3. Initial Considerations and Research Questions

For this analysis, following a review of the driver surveys, initial testing, study objectives from NHTSA, and results from the FMEA, a comprehensive list of proposed research questions was developed by the study team. This list, found in Table 2, contains questions that summarize the overarching themes found to be of concern by drivers. These questions are grouped into six categories, which highlight key technical issues for testing and issues relevant to human-subject testing.

**Table 2.** Proposed research questions for further CBMS testing.

| Group 1—Perceptions of Safety | |
|---|---|
| What procedures exist and how do they appropriately address known safety issues for North American drivers, such as unit magnification and safe gap acceptance judgments? | How can it be demonstrated that replacing outside rearview mirrors with camera monitor systems will not negatively affect light- and heavy-vehicle driver behavior, situational awareness while driving, and overall safety in the United States? |
| What assessment can be performed for the anticipated rates of failure of these camera monitor systems, the anticipated cost to replace them, and the anticipated consumer willingness to replace them? | Does lane-change performance differ when driving with the tested CBMS compared with original equipment (OE) outside rearview mirrors? |
| Do drivers' subjective impressions of general use, comfort, and visibility differ for the tested CBMS compared with OE outside rearview mirrors? | Are there additional tests to observe the efficacy of the anti-glare feature using film coating? |
| What additional tests can be performed to ensure that the arm/restraint to hold the mirror in place is secure in multiple operation design domains (ODDs)? | |
| **Group 2—Response latency** | |
| Are there any potential disadvantages, such as increased eye glance durations, that currently exist for wide-view images? | Does driving with the tested CBMS result in differences in distance judgments when passing a slower lead vehicle compared with OE outside rearview mirrors? |
| Does driver eye-gaze behavior during lane-change and passing maneuvers differ with the tested CBMS compared with OE outside rearview mirrors? | Does driver head movement during lane-change and passing maneuvers differ with the tested CBMS compared with OE outside rearview mirrors? |
| What performance test procedures evaluate image blooming and other display aberrations to ensure that image quality is sufficient to support safe driving? | To the extent that procedures that appropriately evaluate display aberrations do not exist, what new procedures need to be developed? |
| How well does the mirror reflect and magnify the rear image during forward driving? | How well does the mirror reflect and magnify the rear image during a stationary power off or while backing up? |
| **Group 3—Health and fatigue effects** | |
| What test procedures can be developed to measure the impacts of monitor glare from sunlight and other vehicles' headlights, which may cause safety issues? | Does illumination from the tested CBMS's visual displays hinder drivers' ability to detect forward obstacles in darkness? |
| What validation exists so that the petitioned performance specifications and test procedures for ensuring adequate real-world weatherproof performance of side-mounted cameras are adequate? | Does a 360° view create any issues or add to fatigue? |
| What research or information demonstrates that CBMSs will not introduce driver confusion or other driver performance issues among various driver demographics? | |
| **Group 4—Effects of overstimulation** | |
| Although blind spot removal can potentially be achieved with a 360° view for the driver, what additional tests should be performed to ensure that this is true in different ODD scenarios? | Are there additional tests to be performed to ensure that the mirror angle is set and maintained appropriately for clear rear-view visibility under multiple ODDs? |
| **Group 5—Failure recovery** | |
| What tests have been performed to judge the appropriate power-off scenario to ensure safe exit of the vehicle and assurance of safe surroundings? | What performance criteria need to be developed to establish the appropriate minimum residual power duration and test procedures for confirming systems meet the specified duration? |
| How can the heating/defrosting function be performed during a malfunction? | What tests can be performed to ensure safe merging in the case of a system malfunction? |
| What test procedures can be developed to evaluate the anticipated failure rates for such displays when used continuously? | |
| **Group 6—Lighting** | |
| What are the potential delays in the anti-glare feature when entering or exiting a tunnel? | What is the potential for image distortion or processing delays because of rapid changes in lighting (due to sunlight reflecting off objects or entering the field of view through narrowly spaced trees)? |
| What are the potential effects of image distortion or an anti-glare malfunction due to atypical headlights from other vehicles (e.g., blue, or green headlights)? | What are the potential effects of image distortion due to reflection off large bodies of water? |
| What are the potential effects of image distortion from strobing lights (from vehicles or signage)? | |

## 3. Standards

Prior to further testing and research comparing standard glass mirrors with CBMSs, current standards must be reviewed to determine applicability in varying scenarios and determine relevance in testing. In the following section, standards for standard glass (conventional) mirrors and CBMSs are presented in detail.

*3.1. Relevant Standards for Conventional Mirror Systems*

- Federal Motor Vehicle Safety Standard (FMVSS) No. 111: "Rearview mirrors" [9]: This standard specifies the requirements for rear visibility devices and systems, including their performance and location. In particular, the standard defines the required field of view and mounting for inside and outside rearview mirrors. The specifications apply to passenger cars, multipurpose passenger vehicles, trucks, buses, school buses, and motorcycles. This standard is crucial in testing because it provides the requirements for field of view needed.

- ISO 5740: *Road vehicles—Rear view mirrors—Test method for determining reflectance* [10]: This standard specifies a test method for determining the reflectance of rearview mirrors (i.e., the ratio of the luminous flux reflected by the mirror to the luminous flux that the mirror receives as input). It applies to flat mirrors and to mirrors that have convex surfaces for external or internal mounting. This standard is important in judging the mirrors' ability to provide an image to the user but also to project glare.

*3.2. Relevant Standards for Camera-Based Mirror Systems*

- ISO 16505: *Road vehicles—Ergonomic and performance aspects of Camera Monitor Systems—Requirements and test procedures* [11]: This standard summarizes the minimum safety, ergonomics, and technical requirements to replace mandatory mirrors with CBMSs (referred to as camera monitor systems [CMS] in the standard). It specifically refers to classes I to IV as defined in the United Nations' (UN) Regulation No. 46. The specifications are provided using a system-level approach that is applicable to any technology for the camera, display, and processing devices. The standard also includes test procedures related to operational readiness, field of view, magnification, object size and resolution of the CMS, monitor integration inside the vehicle, failure behavior, and influences from the weather and environment, among others. The outlined requirements are based on the properties of the mirror system to be replaced and aspects such as the visual acuity of the human operator. Some relevant definitions provided in the standard include the following: driver's ocular points, ocular reference point, mirror distance to driver's ocular reference point including maximum and minimum values, mirror viewing angle, angle of curvature, field of view for the mirror and camera, magnification factor, visual acuity of the human eye, camera resolution, monitor coordinate system, monitor viewing distance, monitor designed size, and CMS magnification factor. One key assumption in this standard is that the CMS can generate an ideal mapping of the real-world environment.

- ISO 9241: *Ergonomics of human–system interaction* [12]:

  (a) Part 302: "Terminology for electronic visual displays": This part of the standard includes a list of definitions related to electronic visual displays and those used in other sections of ISO 9241. Some of the relevant definitions included in this standard are image formation time, monitor coordinate system, and luminance contrast.

  (b) Part 305: "Optical laboratory test methods for electronic visual displays": This part of the standard defines methods for optical tests and observations used to predict the display's performance considering the ergonomic aspects included in part 303.

  (c) Part 307: "Analysis and compliance test methods for electronic visual displays": This part of the standard focuses on how hardware and software can improve the human–machine interface. It includes an overview of requirements to ensure that design principles for computer-based interactive systems are human centered. The details included allow planners and project managers to understand the relevance of involving technical human factors and ergonomics in the design process. The standard also establishes test methods to analyze visual display technologies, tasks, and environments independently of the technology. Definitions for reality in testing and visual artifacts are also included. The

human factors, ergonomics, and accessibility aspects are treated in more detail in other standards (ISO 6385 and some parts of ISO 7241).

- ISO 15008: *Road vehicles—ergonomic aspects of transport information and control systems—specifications and test procedures for in-vehicle visual presentation* [13]: This standard focuses on the on-board transport information and control systems that are used while the vehicle is on a driving mission. In particular, it defines the requirements for image quality and legibility of displays that show dynamic visual information to the driver independently of the technology. It also includes methods and measurements to assess compliance. This standard is relevant for monitor integration and defines the meaning of luminance, contrast rendering, and image quality measurement.

- ISO 4513: *Road vehicles—Visibility—Method for establishment of eyellipses for driver's eye location* [14]: This document establishes procedures to determine the location of the eyes of the driver inside a vehicle using elliptical 3D models. The definition of neck pivot points is given to define left and right eye points for viewing tasks such as those included in SAE J1050. This standard applies to class A and B vehicles.

- ISO 26262: *Road vehicles—Functional safety—Part 1: Vocabulary* [15]: This standard is applicable to safety-related systems that use one or more electrical/electronic (E/E) systems if they are part of road vehicles. Mopeds are excluded. It is focused on potential hazards due to malfunctioning behavior of E/E systems related to safety functions in vehicles. The following hazards are excluded unless they are directly related to malfunctioning behavior of the E/E systems: electric shock, fire, smoke, heat, radiation, toxicity, flammability, reactivity, corrosion, and release of energy. To help with the integration of functional safety activities in the developmental framework of a company, the document describes a complete framework for functional safety. The requirements included in the standard focus on (1) technical aspects of the functional safety implementation of a product or (2) the developmental process to demonstrate the functional safety capabilities of the organization.

- ISO 12233: *Photography—electronic still picture imaging—resolution and spatial frequency responses* [16]: This standard defines methods to measure the resolutions and spatial frequency responses of electronic still-picture cameras. It applies to monochrome and color cameras that produce digital data or analogue video signals.

- ISO 5385: *Road vehicles—Anti-fog coating for exterior lighting devices—Specification* [17]: The methods and requirements for anti-fog coatings of the exterior lighting devices of road vehicles are specified in this standard. It defines an anti-fog coating as a coating that can prevent all fogging on the exterior of a lens that has been made of glass, polycarbonate, polymethyl methacrylate, or other materials. An exterior lightning device is located outside of a vehicle and performs an illuminating or signaling function.

- ISO 2813: *Paints and Varnishes- Determination of gloss value at 20°, 60° and 85°* [18]: This standard applies to the monitor housing's gloss and describes a method to determine the gloss value of coatings that is applicable to the measurement of gloss in nontextured coatings for plane and opaque substrates.

- UN Regulation No. 46: *Uniform provisions concerning the approval of devices for indirect vision and of motor vehicles with regard to the installation of these devices* [19]: This regulation applies to devices for indirect vision and their installation on category M and N motor vehicles and on motor vehicles with less than four wheels that enclose, partially or wholly, the driver. It lists relevant definitions such as devices for indirect vision, mirror, surveillance mirror, principal radii of curvature at one point on the reflecting surface, spherical surface, aspherical surface, and aspherical mirror, among others. It also outlines the steps that need to be followed to apply for the approval of devices for indirect vision including required information. Requirements and general specifications for mirrors and camera monitor devices for indirect vision are included along with the test procedures that should be performed. This regulation introduces some modifications to parameters defined in ISO 16505.

## 4. Previous Testing Found in the Literature

In this analysis, an overview is provided of 16 recent studies on the performance of CBMSs compared with traditional mirrors in vehicles and on different configurations of a CBMS. A summary is also provided in Table 3 of the major metrics used in experiments and surveys that quantified the usage and design of CBMSs in vehicles.

**Table 3.** Major metrics used for quantifying the performance of CBMSs.

| Metric | # of Papers | [20] | [21] | [22] | [23] | [24] | [25] | [26] | [27] | [28] | [29] | [30] | [31] | [7] | [32] | [33] | [8] |
|---|---|---|---|---|---|---|---|---|---|---|---|---|---|---|---|---|---|
| Subjective questionnaire/survey | 11 | | | | X | X | | X | X | X | X | X | | X | X | X | X |
| Eye-off-road time | 5 | | X | X | | | | | | X | X | X | | | | | |
| Mental workload | 5 | X | X | | | | | | | X | X | X | | | | | |
| Number of off-road glances | 4 | | | X | | | | | | | X | | X | | | | X |
| Eye movement | 4 | X | | | X | | X | | | | | | | | | X | |
| Off-road glance duration | 3 | | | X | | | | | | | X | | X | | | | |
| Driving behavior (e.g., speed variability) | 3 | X | | | | | | | | | | | X | | X | | |
| Distance perception | 2 | | | | | | | | | | | | | | X | | X |
| Driver response time | 2 | | | | | | | | | X | | | | | X | | |
| Yaw angle of neck | 2 | | | | X | | | | | | | | | | X | | |
| Field of view | 1 | | | | | | | | | | | | | X | | | |
| Number of over-the-shoulder checks | 1 | | | | | | | | | | | | X | | | | |
| Cognitive load (oxygenated hemoglobin level) | 1 | X | | | | | | | | | | | | | | | |
| Driving performance | 1 | X | | | | | | | | | | | | | | | |

In reference to the findings in Table 3, one of the most important factors affecting the performance of CBMSs is their ability to capture high-quality images in a variety of lighting conditions. Several studies have evaluated the performance of these systems under different lighting conditions and found that they can produce high-quality images even in low-light conditions [7]. However, the quality of the images captured with these systems can be affected by various factors, such as the quality of the cameras, the positioning of the cameras, and the processing algorithms used to generate the images [29]. In addition, CBMSs provide an accurate and comprehensive view of the surrounding environment. Studies have shown that these systems can provide wider and more comprehensive fields of view than traditional mirrors, which can improve visibility and reduce blind spots [21,28,30]. Additionally, these systems can provide features such as distance measurement, object detection, and warning systems that can improve safety and reduce the risk of accidents [21,22]. Moreover, depending on the design, CBMSs may be more aerodynamically efficient than traditional mirrors, which can improve fuel efficiency and reduce emissions [23,24,31]. Furthermore, these systems can be more compact and less obtrusive than traditional mirrors, which can improve the aesthetic appeal of vehicles and reduce wind noise [24].

Table 3 indicates that several important considerations are lacking in CBMS research. The field of view, number of over-the-shoulder checks, cognitive load (oxygenated hemoglobin level), and driving performance were addressed, but compared with other metrics, were not commonly used. These metrics are related to driver fatigue and overall system performance and should be considered in future work to assess mental load and potential for overstimulation of the driver by CBMSs.

Table 4 identifies whether the literature addresses the key research questions developed in the present work and, if so, whether these questions are addressed qualitatively, quantitatively, or both.

**Table 4.** Key research questions addressed in the literature.

| Questions | [20] | [21] | [22] | [23] | [24] | [25] | [26] | [27] | [28] | [29] | [30] | [31] | [7] | [32] | [33] | [8] |
|---|---|---|---|---|---|---|---|---|---|---|---|---|---|---|---|---|
| Group 1—Perceptions of safety | | | | | | | | | | | | | | | | |
| What procedures exist and how do they appropriately address known safety issues for North American drivers, such as unit magnification and safe gap acceptance judgments? | | | | | | | | | | | | | | | | |
| How can it be demonstrated that replacing outside rearview mirrors with camera monitor systems will not negatively affect light- and heavy-vehicle driver behavior, situational awareness while driving, and overall safety in the United States? | | | O | Y | | | Y | Y | X | O | Y | | Y | O | | O |
| What assessment can be performed for the anticipated rates of failure of these camera monitor systems, the anticipated cost to replace them, and the anticipated consumer willingness to replace them? | | | | Y | | | Y | Y | X | | | | | Y | | |
| Does lane-change performance differ when driving with the tested CBMS compared with OE outside rearview mirrors? | | | | X | | | | | | X | X | | | | | |
| Do drivers' subjective impressions of general use, comfort, and visibility differ for the tested CBMS compared with OE outside rearview mirrors? | | | | Y | | | Y | Y | Y | Y | Y | | | Y | | Y |
| Are there additional tests to observe the efficacy of the anti-glare feature using film coating? | | | | | | | | | | | | | | | | |
| What additional tests can be performed to ensure that the arm/restraint to hold the mirror in place is secure in multiple ODDs? | | | | | | | | | | | | | | | | |
| Group 2—Response latency | | | | | | | | | | | | | | | | |
| Are there any potential disadvantages, such as increased eye glance durations, that currently exist for wide-view images? | X | X | X | | X | | | | X | X | X | X | | | X | X |
| Does driving with the tested CBMS result in differences in distance judgments when passing a slower lead vehicle compared with OE outside rearview mirrors? | | | | | | | | | | | X | | | | X | X |
| Does driver eye-gaze behavior during lane-change and passing maneuvers differ with the tested CBMS compared with OE outside rearview mirrors? | | | | X | | | | | | X | X | | | | | X |
| Does driver head movement during lane-change and passing maneuvers differ with the tested CBMS compared with OE outside rearview mirrors? | | | | X | | | | | | | | | | | | |
| What performance test procedures evaluate image blooming and other display aberrations to ensure that image quality is sufficient to support safe driving? | | | | | | | | | | | | | | | | |
| To the extent that procedures that appropriately evaluate display aberrations do not exist, what new procedures need to be developed? | | | | | | | | | | | | | | | | |
| How well does the mirror reflect and magnify the rear image during forward driving? | | | | | | | | | | | | | | | | |
| How well does the mirror reflect and magnify the rear image during a stationary power off or while backing up? | | | | | | | | | | | | | | | | |
| Group 3—Health and fatigue effects | | | | | | | | | | | | | | | | |
| What test procedures can be developed to measure the impacts of monitor glare from sunlight and other vehicles' headlights, which may cause safety issues? | | | | | | | | | | | | | | | | |
| Does illumination from the tested CBMS's visual displays hinder drivers' ability to detect forward obstacles in darkness? | | | | | | | | | | | | | | | | |
| What validation exists so that the petitioned performance specifications and test procedures for ensuring adequate real-world weatherproof performance of side-mounted cameras are adequate? | | | | | | | | | | | | | | | | |

**Table 4.** *Cont.*

| Questions | [20] | [21] | [22] | [23] | [24] | [25] | [26] | [27] | [28] | [29] | [30] | [31] | [7] | [32] | [33] | [8] |
|---|---|---|---|---|---|---|---|---|---|---|---|---|---|---|---|---|
| Does a 360° view create any issues or add to fatigue? What research or information demonstrates that CBMS will not introduce driver confusion or other driver performance issues among various driver demographics? | X | X | | | | | | | | X | X | | | | | |
| *Group 4—Effects of overstimulation* | | | | | | | | | | | | | | | | |
| Although blind spot removal can potentially be achieved with a 360° view for the driver, what additional tests should be performed to ensure that this is true in different ODD scenarios? Are there additional tests to be performed to ensure that the mirror angle is set and maintained appropriately for clear rearview visibility under multiple ODDs? | | | | | | | | | | | | | | | | |
| *Group 5—Failure recovery* | | | | | | | | | | | | | | | | |
| What tests have been performed to judge the appropriate power-off scenario to ensure safe exit of the vehicle and assurance of safe surroundings? What performance criteria need to be developed to establish the appropriate minimum residual power duration and test procedures for confirming systems meet the specified duration? How can the heating/defrosting function be performed during a malfunction? What tests can be performed to ensure safe merging in the case of a system malfunction? What test procedures can be developed to evaluate the anticipated failure rates for such displays when used continuously? What are the system durability, critical failure mode, expected maintenance schedule, and related procedures? | | | | | | | | | | | | | | | | |
| *Group 6—Lighting* | | | | | | | | | | | | | | | | |
| What are the potential delays in the anti-glare feature when entering or exiting a tunnel? What is the potential for image distortion or processing delays because of rapid changes in lighting (due to sunlight reflecting off of objects or entering the field of view through narrowly spaced trees)? What are the potential effects of image distortion or an anti-glare malfunction due to atypical headlights from other vehicles (e.g., blue or green headlights)? What are the potential effects of image distortion due to reflection off large bodies of water? What are the potential effects of image distortion from strobing lights (from vehicles or signage)? | | | | | | | | | | | | | | | | |

Note: X indicates only quantitative, Y indicates only qualitative, and O indicates both quantitative and qualitative.

According to the findings presented in Table 4, the following gaps exist in the literature and should be addressed in future research:

- Gaps in group 1—perceptions of safety: unit magnification in helping judge gap distances, anti-glare efficacy, and secured restraint of the mirror/camera system. These issues pertain to secondary functions of the CBMS. Testing scenarios both on road and in a simulation will help answer these questions.
- Gaps in group 2—response latency: image blooming, display aberrations, and mirror function while the system is powered off. The first two of these issues are secondary functions, whereas the third is a primary function of the system.
- Gaps in groups 3 and 4—health and fatigue issues and overstimulation: effects of glare and illumination of the screen on vision, and fatigue caused by a 360° view. These issues can be addressed in testing with eye tracking and facial expression analysis systems.
- Gaps in group 5—failure recovery: maintenance of the rear view, safe exiting, heating/defrosting, and merging during malfunctions. These issues will likely need to be addressed in future on-road testing.
- Gaps in group 6—lighting: delays in anti-glare due to tunnels or headlights, effects of the sun angle on image processing, and image distortion from reflection off bodies of water or strobing lights. These issues will require both lab and on-road testing scenarios to judge the effectiveness of these features.

## 5. Conclusions and Future Work

Examination of surveys, standards, and the literature on the testing of CBMSs reveal many opportunities for expansion of this research. From reviewing the literature, CBMSs have several advantages over traditional mirrors, including improved visibility, safety features, aerodynamic efficiency, and aesthetic appeal. Several factors can affect CBMS functionality, such as lighting conditions and camera positioning. Therefore, CBMSs are likely to become increasingly popular in the automotive industry as a replacement for traditional mirrors. However, it is imperative that additional testing be performed, as there are many key research questions outlined in this analysis that are not addressed in the current literature. The significance of this analysis is that the results uncover considerations and metrics to include in future testing of CBMSs. These findings can help ensure that future CBMS research includes key metrics to evaluate driver fatigue.

Regarding future development of testing scenarios involving CBMSs and related technology, additional lab and on-road testing involving human subjects is needed, especially to address the key research questions that were excluded in the literature outlined in the previous section (Table 4). These findings also support the need for the development of a model to evaluate the effectiveness of CBMSs in varying environments, which will be of value to fleet managers and other stakeholders. Figures 6 and 7 show that the freight network and resulting freight vehicle miles traveled are expansive, covering many geographical areas with varying environmental conditions. Hence, to ensure robustness of the technology for future wide-scale implementation, there is a need to evaluate CBMSs and related technologies in heterogenous landscapes and under varying climatic scenarios to determine operational effectiveness in adapting to on-road changes in the environment.

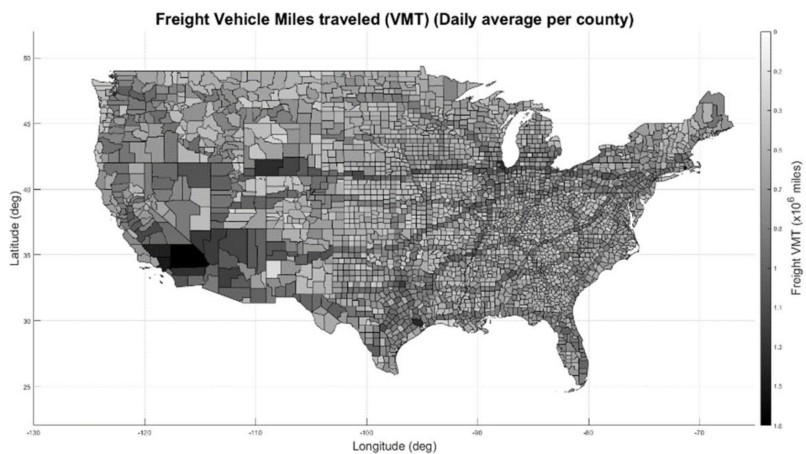

**Figure 6.** Freight vehicle miles traveled (daily average per county).

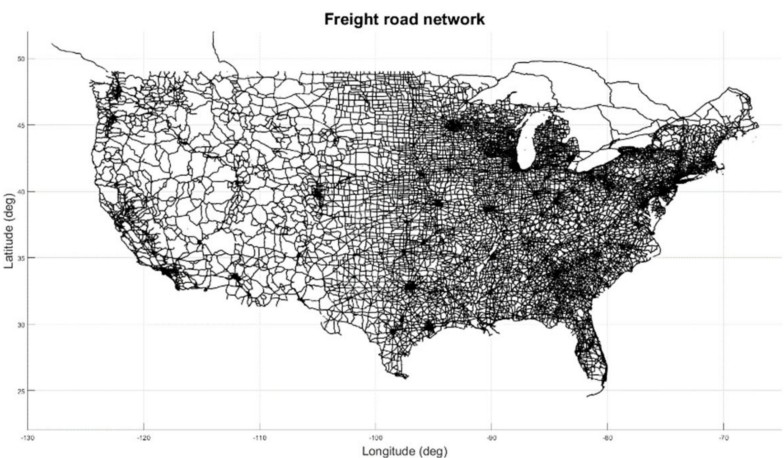

**Figure 7.** Freight network.

**Author Contributions:** Conceptualization, V.S., A.M., A.S., J.R.T., S.O. and J.Y.; Data curation, V.S., A.M., J.R.T., S.O. and J.Y.; Formal analysis, A.M., J.R.T., S.O., J.Y. and V.S.; Funding acquisition, V.S. and A.S.; Investigation, V.S., A.S., A.M., J.R.T., S.O. and J.Y.; Methodology, V.S., A.M., J.R.T., S.O. and J.Y.; Project administration, V.S.; Resources, V.S. and A.S.; Software, V.S. and A.M.; Supervision, V.S.; Validation, V.S., A.M., J.R.T., S.O. and J.Y.; Visualization, V.S., A.M., S.O. and J.Y.; Writing—original draft, V.S., A.M., J.R.T., S.O. and J.Y.; Writing—review and editing, V.S. and A.M. All authors have read and agreed to the published version of the manuscript.

**Funding:** This manuscript has been authored by UT-Battelle, LLC, under contract DE-AC05-00OR22725 with the US Department of Energy (DOE). The US government retains and the publisher, by accepting the article for publication, acknowledges that the US government retains a nonexclusive, paid-up, irrevocable, worldwide license to publish or reproduce the published form of this manuscript, or allow others to do so, for US government purposes. DOE will provide public access to these results of federally sponsored research in accordance with the DOE Public Access Plan (https://www.energy.gov/doe-public-access-plan, accessed on 14 May 2023).

**Institutional Review Board Statement:** Not applicable.

**Informed Consent Statement:** Not applicable.

**Data Availability Statement:** The data presented in this study are available on request from the corresponding author. All data generated and used in this study are summarized and referenced in the text of the document.

**Conflicts of Interest:** The authors declare no conflict of interest.

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
