# Peer review of "Key Considerations in Assessing the Safety and Performance of Camera-Based Mirror Systems"

_safety, 2023_

Round 1

Reviewer 1 Report

The paper presents a very interesting and current topic. However, significant improvements are needed to make this manuscript a satisfactory scientific paper.

In addition to technical deficiencies, it is necessary to arrange the structure of the work itself and improve each chapter (introduction, literature review, methodology, results…).

Author Response

Reviewer 1:

The paper presents a very interesting and current topic. However, significant improvements are needed to make this manuscript a satisfactory scientific paper.

In addition to technical deficiencies, it is necessary to arrange the structure of the work itself and improve each chapter (introduction, literature review, methodology, results…).

Response: The structure of the paper has been improved per our professional editor’s recommendations. Our editor did not advise us to change the order of the sections because we are not developing a model and we do not have formal results to present in the paper.

Reviewer 2 Report

In this manuscript, the authors have studied/surveyed about the implementation of Camera-based mirror systems (CBMS) in place of standard mirrors on trucks from the safety perspective.

The introduction is well written by considering relevant literature in the domain.

Are these research questions in table 2 are prepared by authors (or) adapted from prior literature? What is the authors contributions in questioner.

List the assumptions and limitations of this survey.

Please avoid the detailed descriptions of standard (which is available in the open literature)

Please do any comparative assessment with similar studies to draw the merits of the current work.

Author Response

Reviewer 2:

In this manuscript, the authors have studied/surveyed about the implementation of Camera-based mirror systems (CBMS) in place of standard mirrors on trucks from the safety perspective.

The introduction is well written by considering relevant literature in the domain.

Response: Thank you.

Are these research questions in table 2 are prepared by authors (or) adapted from prior literature? What is the authors contributions in questioner.

Response: The research questions were developed by the research team based on the findings from the driver surveys, NHTSA studies, and the FMEA. The source and explanation of these questions have been added to the paragraph above Table 2.

List the assumptions and limitations of this survey.

Response: The assumptions and limitations have been included in section 1.

Please avoid the detailed descriptions of standard (which is available in the open literature)

Response: The summary of the relevant standards was provided to the readers as a quick overview (to avoid scouring the literature).

Please do any comparative assessment with similar studies to draw the merits of the current work.

Response: Additional studies have been added to compare methodologies.

Reviewer 3 Report

1.      Please change the format of the paper. There is no space at the beginning of the sentence in line 27, and there is more space at the beginning of the sentence in line 98.

2.      There is also a formatting problem with line 191.

3.      The categorical values used in FMEA may not accurately represent the true likelihood and severity of failures. The use of color coding (e.g., green, yellow, orange, red) to represent different risk levels can oversimplify the complexity of failure scenarios and may not provide a precise assessment of the risks involved.

4.      Please check all subheadings of the article, line 146 should be numbered 1.3.

5.      The text mentions that CBMS may be more aerodynamically efficient, compact, and less obtrusive than traditional mirrors, but it does not discuss any potential trade-offs or limitations associated with these advantages. It is important to address potential drawbacks or challenges related to the use of CBMS, such as cost, reliability, maintenance, or potential system failures.

6.      Line 136. In Error! There is a problem of irregular expression, which is very abrupt.

7.      The followings studies were recommended to be properly cited: [1] Probabilistic models of freeway safety performance using traffic flow data as predictors.Safety Science, 2008, 46(9):1306-1333. [2]AI-Empowered Speed Extraction via Port-Like Videos for Vehicular Trajectory Analysis, IEEE Transactions on Intelligent Transportation Systems, vol. 24, no. 4, pp. 4541-4552, 2023.

English can be improved.

Author Response

Reviewer 3:

  1. Please change the format of the paper. There is no space at the beginning of the sentence in line 27, and there is more space at the beginning of the sentence in line 98.

Response: Formatting errors have been addressed by our professional editor.

  1. There is also a formatting problem with line 191.

Response: Formatting errors have been addressed by our professional editor.

  1. The categorical values used in FMEA may not accurately represent the true likelihood and severity of failures. The use of color coding (e.g., green, yellow, orange, red) to represent different risk levels can oversimplify the complexity of failure scenarios and may not provide a precise assessment of the risks involved.

Response: The scoring and explanation of the FMEA have been expanded.

  1. Please check all subheadings of the article, line 146 should be numbered 1.3.

Response: Formatting errors have been addressed by our professional editor.

  1. The text mentions that CBMS may be more aerodynamically efficient, compact, and less obtrusive than traditional mirrors, but it does not discuss any potential trade-offs or limitations associated with these advantages. It is important to address potential drawbacks or challenges related to the use of CBMS, such as cost, reliability, maintenance, or potential system failures.

Response: Thank you for this suggestion. The drawbacks have been included in section 1.

  1. Line 136. In Error! There is a problem of irregular expression, which is very abrupt.

Response: Formatting errors have been addressed by our professional editor.

  1. The followings studies were recommended to be properly cited: [1] Probabilistic models of freeway safety performance using traffic flow data as predictors. Safety Science, 2008, 46(9):1306-1333. [2]AI-Empowered Speed Extraction via Port-Like Videos for Vehicular Trajectory Analysis, IEEE Transactions on Intelligent Transportation Systems, vol. 24, no. 4, pp. 4541-4552, 2023.

Response: Thank you for these source recommendations. They were not included in this paper, but the IEEE ITS paper will be cited in a paper that is forthcoming regarding port maintenance.

Reviewer 4 Report

1.      In line 45 of the article, the serial number of the picture was incorrectly labeled.

2.      The text in pictures 3 and 4 was not clear enough.

3.      The picture needs to be drawn out in the paragraph, such as images 3, 4, and 5.

4.      Note the problem with chapter numbering, such as line 146 in the article.

5.      A comprehensive summary and recommendation was missing after Table 3.

6.      The article lacks innovation and a clear structure. Please reorganize the content to provide a logical flow of ideas and concepts in the chapter 2. Improve the clarity of the language and ensure the writing style is consistent throughout the manuscript.

7.      Strengthen the discussion by addressing the significance of the results and their potential impact on the field.

Author Response

Reviewer 4:

  1. In line 45 of the article, the serial number of the picture was incorrectly labeled.

Response: Formatting errors have been addressed by our professional editor.

  1. The text in pictures 3 and 4 was not clear enough.

Response: Formatting errors have been addressed by our professional editor.

  1. The picture needs to be drawn out in the paragraph, such as images 3, 4, and 5.

Response: Formatting errors have been addressed by our professional editor.

  1. Note the problem with chapter numbering, such as line 146 in the article.

Response: Formatting errors have been addressed by our professional editor.

  1. A comprehensive summary and recommendation was missing after Table 3.

Response: This information was added to the section following Table 3.

  1. The article lacks innovation and a clear structure. Please reorganize the content to provide a logical flow of ideas and concepts in the chapter 2. Improve the clarity of the language and ensure the writing style is consistent throughout the manuscript.

Response: The structure of the paper has been improved per our professional editor’s recommendations.

  1. Strengthen the discussion by addressing the significance of the results and their potential impact on the field.

Response: Significance and impact have been added to the discussion section.

Reviewer 5 Report

The paper suggests an important problem, however, it fails to provide a good solution.

the technical writing is very bad it is hard to read or consider.

the paper cannot be accepted in this format

The writing is very bad and hard to follow

Author Response

Reviewer 5:

The paper suggests an important problem, however, it fails to provide a good solution.

the technical writing is very bad it is hard to read or consider.

the paper cannot be accepted in this format

Comments on the Quality of English Language

The writing is very bad and hard to follow

Response: The paper has been reviewed and edited by a professional editor. The necessary changes have been made to reflect potential research focus and direction for future studies.

Reviewer 6 Report

Review

Key Considerations in Assessing the Safety and Performance of Camera-Based Mirror Systems

Amy Moore, Jinghui Yuan, Shawn Ou, Jackeline Rios Torres, Vivek Sujan, Adam Siekmann

Subject

The research topic is interesting and appropriate for the journal requirements.

Title

Clear and corresponds to the content of the article.

Abstract

Appropriate, including the purpose of the research. The abstract is clear, arouses interest, and consists of the goals.

1. Introduction n to camera-based mirror systems

The description is sufficiently detailed and understandable. The tables and figures fit well with the content.

2. Findings from the literature

The number of the earlier studies presented is good as there are previously known scientific publications on the subject.

3. Conclusion and future work

The conclusions are appropriate based on the results presented.

References

It contains the most important scientific publications, but it could be more extensive in terms of quantity

Overall merit

The subject is essential. This paper is well structured and sufficiently detailed. The descriptions are understandable; the results are well presented and evaluated.

Author Response

Reviewer 6:

Key Considerations in Assessing the Safety and Performance of Camera-Based Mirror Systems

Amy Moore, Jinghui Yuan, Shawn Ou, Jackeline Rios Torres, Vivek Sujan, Adam Siekmann

Subject

The research topic is interesting and appropriate for the journal requirements.

Title

Clear and corresponds to the content of the article.

Abstract

Appropriate, including the purpose of the research. The abstract is clear, arouses interest, and consists of the goals.

  1. Introduction n to camera-based mirror systems

The description is sufficiently detailed and understandable. The tables and figures fit well with the content.

  1. Findings from the literature

The number of the earlier studies presented is good as there are previously known scientific publications on the subject.

  1. Conclusion and future work

The conclusions are appropriate based on the results presented.

References

It contains the most important scientific publications, but it could be more extensive in terms of quantity

Overall merit

The subject is essential. This paper is well structured and sufficiently detailed. The descriptions are understandable; the results are well presented and evaluated.

Response: Thank you so much. Additional studies have been added to compare methodologies.

Round 2

Reviewer 1 Report

Dear authors,

Additional effort is needed to improve the manuscript. As emphasized in previous comments, it is necessary to improve each chapter.

Reviewer 2 Report

the manuscript is considerably modified, and no further queries from my side.

All the best.

Reviewer 3 Report

Comments were not addressed.

English should be addressed.

Reviewer 4 Report

No more comments.

Reviewer 5 Report

The paper has been significantly improved.

It can be considered for acceptance in this format.

The written English is acceptable.